# CorDial: Coarse-to-fine Abstractive Dialogue Summarization with Controllable Granularity

## Abstract

Dialogue summarization is challenging due to its multi-speaker standpoints, casual spoken language style, and limited labeled data. In this paper, we propose CorDial, aiming to improve the abstractive dialogue summarization quality and, at the same time, enable granularity controllability. We propose 1) a coarse-to-fine generation strategy that generates a summary draft followed by a final summary. The summary draft, which provides weakly-supervised signals, comprises pseudo-labeled interrogative pronoun categories and noisy key phrases extracted with a constituency parser. 2) A simple strategy to control the granularity of the final summary. CorDial can automatically determine or control the number of generated summary sentences for a given dialogue by predicting and highlighting different text spans from the source text. Our model achieves state-of-the-art performance on the largest dialogue summarization corpus SAMSum. We conduct comprehensive case study and show competitive human evaluation results and controllability to human-annotated summaries.

## 1 Introduction

Text summarization tasks aim to distill the most critical information in a text to produce an abridged version. In particular, abstractive – as opposed to extractive – summarization requires neural generative models with a high level of semantic understanding, as the output words do not necessarily appear in the source text. It is more challenging but gives much flexibility to a summary compared to any extractive summarization models (Zhang et al., 2018). Abstractive dialogue summarization has been discussed in the literature of AMI meeting corpus (McCowan et al., 2005). The size and quality of labeled data are bottlenecks, as collecting summaries is costly and judgements when creating them are inherently subjective. The AMI corpus has only 141 summaries, and the largest dialogue summarization dataset SAMSum (Gliwa et al., 2019) has number of training samples only equal to 5% of the commonly-used text summarization dataset CNN/DailyMail (Hermann et al., 2015).

In addition to (and perhaps due to) the shortage of labeled data, dialogue summarization has not received much attention despite the prevalence of dialogues (text messages, emails, social media) and the vast application potential of dialogue summarization systems. Significant research efforts have been focused on summarization of single-speaker documents such as News (Hermann et al., 2015; Nallapati et al., 2016; See et al., 2017) or scientific publications (Qazvinian & Radev, 2008; Nikolov et al., 2018). However, summarizing a dialogue presents a unique set of challenges. A conversation always involves multiple speakers with different points of view, and its natural language style is very different from a standard writing style. For example, conversational data contains more abbreviations and typos. Information is more scattered across in a dialogue, compared to articles where usually titles or the first few sentences contain the most salient information.

Recently, the ability to control text summarization in the News domain has been gradually attracting more attention (Fan et al., 2018; Liu et al., 2018), with work focusing on learning length embeddings to control summary lengths. However, the length information is only added during the decoding stage, making the encoding stage less informed, and the overall conditional generation implicit. Saito et al. (2020) instead first explicitly extract "prototype" text span in the desired length and then

| Morgan | `<hl>` Hey gorgeous, what's up? |
|---|---|
| Suzanne | Nothing special, it's just one of many boring days at work. But's better now tho! |
| Morgan | Are you working at all ? |
| Suzanne | I'm trying but you aren't helping me, at all. I'm just taking a well-deserved break `</hl>` |
| Morgan | `<hl>` I miss you Suzie |
| Suzanne | I miss you too Morgan |
| Morgan | Do you feel like going to a concert next week? maroon 5 is playing at the hulu theater at madison square garden .. as it happens , I've got two tickets. do you want to go ? `</hl>` |
| Suzanne | `<hl>` Really? OMG! That's wonderful !. Thank you sweetheart! |
| Morgan | Oh, nothing. I just want you to be happy :) `</hl>` |

| Turn | Intent | Key Phrase |
|---|---|---|
| 1 | what | - |
| 2 | abstain | "s just one of many boring days at work" |
| 3 | confirm | "working at all" |
| 4 | abstain | "m just taking a well-deserved break" |
| 5,6 | abstain | - |
| 7 | confirm | "feel like going to a concert next week", "maroon 5", "is playing at the hulu theater at madison square garden" |
| 8 | why | - |
| 9 | abstain | - |

*Summary: Suzanne is at work and is having a break now. Morgan invites Suzanne to a concert of Maroon 5 which takes place next week at the Hulu Theatre at Madison Square Garden. Suzanne agrees.*

Figure 1: An input and output example for our proposed solution. Given the dialogue on the left hand side, we first construct summary draft with intent and key phrase information for coarse-to-fine generation. Then, we split the dialogue into several pieces by special tokens for model controllability and interpretability.

paraphrase it to the output News summary. However, the retrieve-and-rewrite process is restricted by the extraction quality, leaving its performance limited by extractive solutions' capabilities.

In this paper, we propose CORDIAL, a coarse-to-fine abstractive dialogue summarization model equipped with granularity controllability. Unlike previous methods (Goo & Chen, 2018; Pan et al., 2018) which heavily rely on explicit intent annotations in datasets, we label each dialogue turn with a pre-defined interrogative pronoun category using a weakly-supervised labeling approach. The automatically labeled user intent together with its corresponding key phrase extraction provide weak supervision during summary generation. In addition, we propose a length-controllable generation method specifically for dialogue summarizaiton. We match each summary sentence "linearly" to its corresponding dialogue context and clip it by highlighting tokens. We then train our model to predict where to clip and generate only one sentence for each highlighted dialogue. This strategy enables CORDIAL to generate a summary at different granularity by highlighting arbitrary numbers of text spans from a dialogue and making our model more interpretable.

We build our model on top of BART-xsum (Lewis et al., 2019), which is first pre-trained with unsupervised denoising objectives, and further fine-tuned on the News summarization corpus XSUM (Narayan et al., 2018). In the experimental results, we show that CORDIAL achieves state-of-the-art dialogue summarization performance on several automatic metrics. The main contributions of this work [1] are: 1) We propose a coarse-to-fine strategy that uses artificial summary draft as weak supervision, 2) we introduce a text-span based conditional generation approach to control the granularity of generated dialogue summaries without human-written summaries at different detail levels, and 3) we conduct comprehensive case study and human evaluation to show that CORDIAL can achieve consistent and informative summary, especially for controllable summary, where existing models either cannot do it or do it poorly.

## 2 METHODOLOGY

In this section, we first briefly cover the background of generative language pre-trained models. Then, we introduce our proposed summary draft construction and summary controllability in detail. The proposed solution is generalizable to all the generative language models. We define the conversational history input as $D = \{X_1, X_2, \ldots, X_N\}$, where each $X_i$ has a sequence of words, $N$ is the

---

[1]Our code is released at `www.anonymous.com`

total numbers of dialogue turns, and the input may contain more than two speakers. We intend to generate $M$-sentence dialogue summary $Y = \{Y_1, \ldots, Y_M\}$ that is suppose to be briefer than the overall dialogue history.

## 2.1 GENERATIVE PRE-TRAINED LANGUAGE MODELS

Self-supervised generative language models (Radford et al., 2019; Yang et al., 2019; Dong et al., 2019; Lewis et al., 2019) have achieved remarkable success in many NLP tasks. Instead of solely training on a specific dataset, pre-training on a large-scale text corpus such as Wikipedia and BookCorpus has shown good knowledge transferability. However, in conversational AI, previous works (Wolf et al., 2019b; Wu et al., 2020) argue that there is an intrinsic difference of linguistic patterns between human conversations and writing text. We would like to answer the following question: which generative language model is the best base model for dialogue summarization tasks. In this paper, we investigate four recently proposed models, DialoGPT (Zhang et al., 2019d), UniLM (Dong et al., 2019), PEGASUS (Zhang et al., 2019a), and BART (Lewis et al., 2019). The first one is a generative language model trained on open-domain conversational data, and the other three achieve promising results on several text summarization benchmarks.

## 2.2 DRAFT CONSTRUCTION

For each turn in the dialogue, we create a summary draft containing the speaker's dialogue action category (defined later) together with the most critical key phrases. Our hypothesis is such summary drafts can provide useful weak supervision beneficial to the final summarization task. Unlike task-oriented dialogue systems, which have explicit and annotated intents (e.g., book flight and check account) and actions (e.g., inform and request), dialogue summarization tasks rarely have such labels. Therefore, we define a set of interrogative pronoun categories, and then label each utterance with its category using a weakly-supervised labeling approach (Ratner et al., 2019).[2]

The interrogative pronoun categories are designed to identify functional units (intent) of all utterances, serving as the dialogue's logic. For example, in Figure 1, Morgan asked Suzanne "Do you feel like going to a concert next week?" One can expect that Suzanne will confirm her willingness in the next dialogue turn. We define such dialogue action categories as follows: *WHY:* asks the reason of the state mentioned in the previous turn, e.g., "why" or "why not"; *WHAT:* requests more details about the aforementioned object, the sentence usually starts with "what's" or "what about"; *WHERE:* asks the location of an appointment or event; *WHEN:* asks the time of an appointment or event, e.g. ,"when" or "what time"; *CONFIRM:* asks the other speaker to establish the correctness of certain case, the sentence usually starts with patterns like "are you", "will you" or "has he"; *AB-STAIN:* the utterance does not belong to any of the previous categories. It occurs when speakers continue to state or comment without seeking for more information from the others.

Following this, we define key phrases as essential words or phrases in the annotated summary. We first use a trained constituency parser (Kitaev & Klein, 2018) to parse each dialogue turn and each summary sentence. Then, we identify the longest common sub-sequence with a threshold among dialogue and summary to be the key phrases. Note that not every dialogue turn contains key phrases and the key phrases could be noisy. Overall, we construct a summary draft by concatenating turn index, dialogue action category label, and extracted key phrases within the entire dialogue history into a string, ending with a special token, "TL;DR". Take Figure 1 as an example, the summary draft is "1 what 2 abstain 's just one of ... square garden 8 why 9 abstain TL;DR". We train our model first to generate this summary draft and then generate the final summary in an autoregressive way. We use TL;DR token to distinguish draft and final summary during inference time.

## 2.3 CONTROLLABILITY

The high-level intuition for our solution is that if we can control a generative model only to generate one sentence as output for a partially-highlighted input, we can control the number of output sentences by choosing how to highlight the input. We highlight each dialogue split using the special token pair $< hl >$ and $< /hl >$. For example, in Figure 1, we generate the first summary sentence for the first segment marked in blue color with the highlighting token pair, and the second

---

[2] github.com/snorkel-team/snorkel

and third by green and orange markers, respectively. This way, we can not only gain the summary controllability but also make the generation more interpretable.

The next challenge is, during training, we have to find a mapping between each sentence in a reference summary to its corresponding dialogue split. In other words, how do we know where to insert the highlighting tokens? We do so by training a dialogue-turn-level binary classifier (detailed below) that predicts whether each turn is a cutting point. Our key observation is that sentences within a reference summary usually have a strong temporal dependency, that is, people summarize the dialogue almost "linearly". We use a naive approach that simply find the dialogue split that has the highest ROUGE score to each summary sentence. The cutting point

$$t_m = \arg\max_t \text{SIM}(X_{c_m:t}, Y_m),$$ (1)

where SIM could be any similarity functions (we use ROUGE-1) available, and $c_m$ is the accumulated turn index ($c_1 = 1$ and $c_m = t_{m-1}$) that indicates which part of a dialogue has been covered by a summary sentence. $t_m$ is the cutting point in the dialogue history for the $m$-th summary sentence. Note that for a summary with $M$ sentences, we only need $M - 1$ cutting points. With the pseudo labels ($t_m$) provided by this heuristic, we formulate the dialogue segmentation problem into a binary classification problem. Specifically, we train a classifier $C$, which takes dialogue history as input and predicts whether each dialogue turn is a cutting point. We prefix each dialogue turn with a separation token (e.g., $x_{sep} = $) and take such long sequence as input to the classifier.

$$H = C([x_{sep}, X_1, x_{sep}, X_2, \ldots, x_{sep}, X_N]) \in \mathbb{R}^{N \times d_{emb}},$$
$$\hat{P} = \text{Sigmoid}(W_1(H)) \in \mathbb{R}^N.$$ (2)

The classifier output $H$ is the representations of those separation tokens, and each of them is a $d_{emb}$ dimension vector. $W_1 \in \mathbb{R}^{d_{emb} \times 1}$ is a trainable linear mapping. The $\hat{P}$ is the predicted segment probability that is trained with binary cross-entropy loss. We use a BERT-base model (Devlin et al., 2018) as a classifier $C$ and the $i$-th cutting point is triggered if $\hat{P}_i > 0.5$. This prediction means that our model can automatically determine how many sentences should be generated in the final summary. If no cutting point is triggered, then we will generate a one-sentence summary. If one cutting point is triggered, we will have a two-sentence summary, and so on. Note that the segment classifier does not need to be perfect (e.g., 86% accuracy in our experiment) as the labels contain a certain noise and splitting a dialogue by one turn earlier or later usually does not make a big difference. As a result, we find that the final ROUGE score is quite similar for both "oracle" dialogue splits and predicted dialogue splits.

Finally, we can control the number of output summary sentences by controlling the dialogue split. Specifically, we first decide the expected number of output sentences (e.g., $K$), and then we choose the top $K - 1$ indexes with highest probabilities in segmentation probability $\hat{P}$. We use these $K - 1$ indexes as cutting points to add highlighting tokens. For example, if we want to have a three-sentence summarization, we can split the dialogue into three pieces by choosing the top two highest segment probability. Then, we input the same dialogue three times with different highlighting portions to CORDIAL and obtain one-sentence summary each. We can also generate one-sentence summary by clipping the whole dialogue with one pair of highlighting tokens at the beginning and the end of a dialogue (we call this output CORDIAL-1).

## 2.4 OVERALL GENERATION

CORDIAL follows a standard encoder-decoder framework. During training, we use "oracle" dialogue segmentation to add highlighting tokens for each summary sentence, separately. We take the highlighted dialogue history as input and train our model to generate its corresponding summary draft and a summary sentence. For example, the first summary sentence, we input whole dialogue with added highlighting tokens both in the beginning of the first turn and in the end of the fourth turn, and generate output that contains corresponding summary draft "1 what 2 abstain ... well-deserved break" and the first summary sentence "Suzanne is at work and is having a break now." The entire model is trained on cross-entropy loss for the generated tokens. During inference, we first use the trained segment classifier to predict splitting points, suggesting how many sentences to be generated in the final summary. Then, we use the predicted segmentation to add highlighting tokens into a dialogue. Finally, after generating multiple summary sentences separately, we simply concatenate them to be the final summary.

## 3 EXPERIMENTS

### 3.1 DATASET

We perform experiments on the recently released SAMSum dataset (Gliwa et al., 2019), which to the best of our knowledge is the most comprehensive resource for abstractive dialogue summarization tasks. It contains 16K natural messenger-like dialogues created by linguists fluent in English with manually annotated summaries. Its comprehensiveness is reflected in the following aspects: 1) Unlike previous datasets consisting of only hundreds of dialogue-summary pairs, it has larger data size - 14,732 training pairs, 818 validation pairs, and 819 test pairs; 2) 75% of the conversations are between two interlocutors, the rest are between three or more people; 3) the conversations cover diverse real-life topics, and the summaries are annotated with information about the speakers.

We preprocess the data by the following steps: 1) concatenate adjacent utterances of the same speaker into one utterance; 2) clean the dialogue text by removing hashtags, URLs and Emojis; 3) label each utterance with its corresponding interrogative pronoun category with a weak supervision approach (Ratner et al., 2019); 4) parse each utterance with a constituency parser and find the longest common sub-sequence between the phrases and summary to be the key phrases.

### 3.2 EVALUATION METRICS AND BASELINES

We use the standard ROUGE metric (Lin, 2004) as automatic evaluation metrics, including ROUGE-1, ROUGE-2, and ROUGE-L. Following previous works (Gliwa et al., 2019), we use py-ROUGE [3] library with stemming function. We compare our model with baselines reported in Gliwa et al. (2019): Longest-3 is a commonly-used extractive summarization baseline which takes the top three longest sentences as summary. The pointer generator and Fast abs are RNN-based methods with copy-attention mechanism or policy gradient. The Transformer is a random-initialized self-attention architecture with multi-head attention. The DynamicConv is a lightweight convolutional model that can perform competitively to self-attention. All of these models are not pre-trained.

Besides, we investigate four pre-trained generative language models. DialoGPT is a GPT model pre-trained on open-domain Reddit data. UniLM is pre-trained using three types of language modeling tasks: unidirectional, bidirectional, and sequence-to-sequence prediction on English Wikipedia and BookCorpus. PEGASUS masks important sentences from input and is trained to generate the missing parts, similar to an extractive summary approach. BART is trained by corrupting text with an arbitrary noising function and learning to reconstruct the original text. We use default parameters listed in the respective open-source repositories to fine-tune on the dialogue summarization task. We show the training details in the Appendix.

### 3.3 RESULTS

In Table 1 of ROUGE results, we find that the methods that are pre-trained or with pre-trained embeddings perform better than those that are not. For instance, DynamicConv achieves a 3–4% improvement by adding GPT-2 embeddings. This further confirms the impact of language model pre-training on downstream tasks. Among the pre-trained generative language models examined, PEGASUS and BART are the top two models with ROUGE-1 higher than 50. Surprisingly, DialoGPT, the model pre-trained on conversational data, does not achieve satisfactory results as one might expect.

CORDIAL achieves the highest 50.79% ROUGE-L score. We also conduct an ablation study by removing summary draft generation (BART+Ctrl) or controllability (BART+Draft). In both cases we observe a performance drop, except a slight improvement on ROUGE-1 for BART+Ctrl. This suggests that the drafting step helps generate a more fluent summary even with lower unigram matching. Furthermore, recognizing the limitation of ROUGE scores in their ability to fully capture the resemblance between the generated summary and the reference, in Table 2, we follow Fabbri et al. (2020) to compare model performances with additional metrics, including ROUGE-Word Embedding (Ng & Abrecht, 2015), BERTScore (Zhang et al., 2019b), MoverScore (Zhao et al., 2019), Sentence Mover's Similarity (SMS) (Clark et al., 2019), BLEU (Papineni et al., 2002), and CIDEr (Vedantam

---

[3]pypi.org/project/pyROUGE/

Table 1: Dialogue summarization ROUGE evaluation on the SAMSum test dataset. Results with *
are obtained from Gliwa et al. (2019). CORDIAL achieves the highest ROUGE score.

|  | ROUGE-1 | ROUGE-2 | ROUGE-L |
|---|---|---|---|
| Longest-3* | 32.46 | 10.27 | 29.92 |
| Pointer Generator (See et al., 2017)* | 37.27 | 14.42 | 34.36 |
| Fast Abs RL (Chen & Bansal, 2018)* | 41.03 | 16.93 | 39.05 |
| Transformer (Vaswani et al., 2017)* | 42.37 | 18.44 | 39.27 |
| DynamicConv (Wu et al., 2019)* | 41.07 | 17.11 | 37.27 |
| DynamicConv + GPT-2 emb* | 45.41 | 20.65 | 41.45 |
| DialoGPT (Zhang et al., 2019d) | 39.77 | 16.58 | 38.42 |
| UniLM (Dong et al., 2019) | 47.85 | 24.23 | 46.67 |
| PEGASUS (Zhang et al., 2019a) | 50.50 | 27.23 | 49.32 |
| BART (Lewis et al., 2019) | 51.74 | 26.46 | 48.72 |
| BART + Draft | 51.79 | 26.85 | 49.15 |
| BART + Ctrl | **52.84** | 27.35 | 50.29 |
| CORDIAL | 52.65 | **27.84** | **50.79** |

Table 2: Dialogue summarization evaluation on the SAMSum test dataset with additional recently
introduced metrics that have been applied to both text generation and summarization.

|  | ROUGE_WE | BERTScore | MoverScore | BLEU | CIDEr | SMS |
|---|---|---|---|---|---|---|
| PEGASUS | 0.3562 | 0.5335 | 0.3233 | 17.33 | 1.741 | 0.1608 |
| BART | 0.3606 | 0.5387 | 0.3391 | 17.55 | 1.701 | 0.1401 |
| CORDIAL | **0.3759** | **0.5458** | **0.3539** | **19.58** | **1.981** | **0.1689** |

et al., 2015). As shown in Table 2, CORDIAL consistently outperforms PEGASUS and BART. More
information about these evaluation metrics are shown in the Appendix.

## 3.4 ANALYSIS

### 3.4.1 HUMAN EVALUATION BY CROWDSOURCING

We leverage human judgement to evaluate the generated summaries via crowdsourcing, especially
for granularity-controlled generation, since we do not have human-written reference summaries of
various lengths (number of sentences). We ask workers to rate the summaries in two aspects on a
scale from -1 (worst) to 1 (best): factual consistency and informativeness. *Factual* consistency acts
as a precision measure, assessing whether the information provided in summary contains factual
errors. *Informativeness* is a recall-oriented measure, examining whether critical information in a di-
alogue is mentioned in summary. We also show the length ratio between a summary and a dialogue,
where a lower ratio means a higher compression rate. For the crowdsourcing evaluation, we randomly
select 30 dialogues, each of which is annotated by three workers. [4]

To show the proposed controllable generation's strengthens and quality, we provide two additional
baselines, Longest-1 and BART-1. The longest-1 method is an extractive baseline that outputs the
longest dialogue turn as the final summary. The BART-1 is a strong abstractive baseline where we
train a BART-based summarization model with the number of summary sentences in the training set
as its start-of-sentence token during decoding. Similar to the approach from Liu et al. (2018), we
can use different start-of-sentence tokens to control the BART output.

In general, it is preferable to have a factually consistent and informative summary that is succinct
(low length ratio, high compression rate) at the same time. As shown in the first row of Table 3,
CORDIAL-1 achieves the highest informative score among all generated one-sentence summaries,
indicating the strength of the proposed controllable method in producing succinct yet informative
dialogue summaries. The Longest-1 method has a higher consistent score because its summary is
directly copied from the original dialogue, preventing any factual mistakes. The second row of
Table 3 shows that CORDIAL, when automatically determining the granularity of the summary,
produces summaries that are more succinct (lower length ratio), more factually consistent, and more
informative, compared to the BART model.

---

[4]The prediction file on the test set is provided in the supplementary file.

Table 3: Human evaluation results on test set for both controllable summary and standard summary.

|  | Length Ratio | Consistent (Precision) | Informative (Recall) |
|---|---|---|---|
| Longest-1 | 0.27 | **0.70** | 0.23 |
| BART-1 | **0.16** | 0.50 | 0.16 |
| CORDIAL-1 | 0.19 | 0.50 | **0.49** |
| BART | 0.26 | 0.65 | 0.51 |
| CORDIAL | **0.24** | **0.69** | **0.53** |
| Gold | 0.27 | 0.74 | 0.55 |

Table 4: A test set example with generated summaries.

| | |
|---|---|
| | Kelly: I still haven't received the rent money. Did you check with your bank?
John: Yes. I definitely sent it last week.
Kelly: But I still don't have it. Can you please check that you sent it to the right account.
John: Ok. Give me 5 min.
Kelly: OK
John: I checked and the money did go out of my account last week.
Kelly: What account number did you send it to?
John: 44-1278
Kelly: No wonder! My account number is 44-1279. You sent it to someone else's account.
John: F*ck! D*mn! F*ck!. I'm really sorry!
Kelly: I still need the rent money though.
John: I'm really sorry I'll have to go to the bank tomorrow and ask if they can re-send it to the right account.
Kelly: Thanks ! |
| Longest-1 | John said I'm really sorry I'll have to go to the bank tomorrow and ask if they can re-send it to the right account. |
| BART-1 | Kelly still hasn't received the rent money from John. |
| CORDIAL-1 | John sent the rent money to the wrong account and will have to ask the bank to re-send it to the correct one tomorrow. |
| BART | Kelly still hasn't received the rent money. John sent it to the wrong account number 44-1278. John will go to the bank tomorrow and ask if they can re-send the money to the right account. |
| CORDIAL | John sent the rent money to the wrong account last week. John will go to the bank tomorrow and ask if he can re-send the money to the correct account. |
| Gold | Kelly hasn't received the rent money, because John sent it to the wrong bank account. He will go to the bank to tackle the issue. |

### 3.4.2 CASE STUDY

CORDIAL outperforms the baseline models in both ROUGE scores and human evaluation metrics. We now further inspect its textual quality. In Table 4, we show an example from the SAMSum test set with summaries generated by different models. In this example, CORDIAL and CORDIAL-1 can both produce a near-perfect summary even compared to the human-written reference summary. On the other hand, the summary generated by BART includes overly detailed information (e.g., bank account). We show some more examples in the Appendix and all the predictions (including CORDIAL-1 and CORDIAL-2) in the supplementary file.

We also manually examine 100 summaries generated from CORDIAL against the reference summaries in the test set. Specifically, we analyze each of the three following problematic cases, where summarization models frequently make mistakes, reported by Gliwa et al. (2019), and provide sample summaries in Table 5. 1) *Associating names with actions*: CORDIAL performs well in dealing with speakers' names. It accurately associates "her dad" with "Ann's dad," also "Fiona tries to help her" with "Ann and Fiona." 2) *Extract information about the arrangement after discussion*: Even speakers hesitate about the flower's color to be yellow, pink or red in the middle of the discussion, CORDIAL still correctly determines the right color after several turns. 3) *Decide important information in dialogues*: CORDIAL fails to capture some of the important facts (marked as red) mentioned in reference summary. We conjecture the reason could be that 1) some of the important facts are located in the same part of the highlighted turns, and 2) those information is missed by the key phrase extraction. Simultaneously, we force the model to generate only the most important one under the constraint of controllability. The improvement of CORDIAL on the first two summarization difficulties can be partially attributed to the clear logic in the draft when an input to the model.

## 4 RELATED WORK

**Neural Text Summarization**    There are two main paradigms for text summarization: extractive and abstractive. Inspired by the success of applying seq2seq models on neural machine translation,

Table 5: Case analyses by manually examining CORDIAL generated summaries.

| | Reference Summary | CORDIAL Summary |
|---|---|---|
| Associat names with actions | Lilly will be late.
Gabriel will order pasta with salmon and basil for her. | Lilly will be late for the meeting with Gabriel.
Gabriel will order something for Lilly. |
| | Ann doesn't know what she should give to her dad as a birthday gift.
He's turning 50.
Fiona tries to help her and suggests a paintball match. | It's Ann's dad's 50th birthday.
He's turning 50. Ann and Fiona are planning a surprise birthday party for her dad. |
| Extract information after the discussion | Paul will buy red roses following Cindy's advice. | Paul wants to buy red roses. |
| Decide important information | Rachel's aunt had an accident and she's in hospital now.
She's only bruised.
The perpetrator of the accident is going to pay for the rehabilitation. | Rachel is at the hospital with her aunt, who had an accident.
She's bruised but fine.
She will give her a hug. |
| | Hannah needs Betty's number but Amanda doesn't have it.
She needs to contact Larry. | Amanda can't find Betty's number.
Amanda suggests to text him. |

Rush et al. (2015) and Nallapati et al. (2016) introduce the neural seq2seq model on abstractive text summarization, with an attention-based encoder and a neural language model decoder. To solve the problem of out-of-vocabulary words and to capture salient information in source documents, See et al. (2017) propose a pointer-generator network that copy words from source to target. Many subsequent works (Gehrmann et al., 2018; Paulus et al., 2018) employ this mechanism and further demonstrate its effectiveness with reinforcement learning. Recently, Liu & Lapata (2019) apply BERT on text summarization and propose a general framework for both extractive and abstractive models. Zhang et al. (2019c) pre-train hierarchical document encoder for extractive summarization. Lewis et al. (2019) introduces BART, a denoising autoencoder for pretraining sequence-to-sequence models. BART significantly outperforms the best previous work in terms of ROUGE metrics.

**Dialogue Summarization**   Most research works (Oya et al., 2014; Mehdad et al., 2014; Banerjee et al., 2015) are conducted on the AMI meeting corpus (McCowan et al., 2005). Goo & Chen (2018) propose to use the topic descriptions as reference summaries and use dialogue acts as clear training signals. However, such summaries are very general since they only describe the high-level goals of meetings. Pan et al. (2018) build the Dial2Desc dataset by reversing a visual dialogue task. Liu et al. (2019) collect their dataset from the logs in the DiDi customer service center. However, the dataset is not publicly available. It is restricted to the task-oriented scenario, where one speaker is the user and the other is the customer agent, with limited topics. Recently, Gliwa et al. (2019) introduce the SAMSum corpus, which contains around 16k chat dialogues with manually annotated summaries. It is the first comprehensive abstractive dialogue summarization dataset spanning over various lengths and topics.

**Length-controllable Generation**   The most prevalent method for length control generation is using a special length embedding. Kikuchi et al. (2016) first propose length control for abstractive summarization by using length embedding as an additional input for the LSTM. Fan et al. (2018) train embeddings that correspond to each different output length and prepend that unique length marker at the beginning of the decoder. Liu et al. (2018) incorporates the length embedding into the initial state of a CNN-based decoder. Takase & Okazaki (2019) extends the positional encoding in a Transformer model by considering the remaining length explicitly at each decoding step. Saito et al. (2020) propose to control the summary length with prototype extractor, but the summary rewriting process is largely bounded to the extraction quality. The aforementioned works all focus on structured text summarization(e.g. news document). We are the first to propose generate length-controllable summary on dialogues by highlighting arbitrary numbers of dialogue spans.

## 5   CONCLUSION

The dialogue summarization task is challenging but with vast application potential. We propose CORDIAL, a state-of-the-art dialogue summarization model with granularity controllability. COR-DIAL uses a weakly-labeled summary draft for its coarse-to-fine generation, and text-span conditional generation for a controllable summary. Our model achieves state-of-the-art results on the largest dialogue summarization dataset. We show with human evaluation that our model can generate factually consistent and informative summaries. We also point out several error cases to shed light on future research direction of controllable dialogue summarization.

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

# A  APPENDIX

## A.1

- DialoGPT: `github.com/microsoft/DialoGPT`
- UniLM: `github.com/microsoft/unilm`
- PEGASUS: `github.com/google-research/pegasus`
- BART `huggingface.co/transformers/model_doc/bart.html`

## A.2  TRAINING DETAILS

We use huggingface (Wolf et al., 2019a) implementation to fine-tune a BART model. We use the large version fine-tuned on the XSUM (Narayan et al., 2018) dataset with 12 self-attention encoder and decoder layers. We truncate input dialogue to a maximal length 512 with training batch size 4. We train the model with Adam optimizer (Kingma & Ba, 2014) with 0.1 proportion for linear learning rate warmup. We early stop on validation set ROUGE-1 score, and it is trained for around 40,000 steps on one NVIDIA V100 GPU. During inference, we do beam search decoding with beam size 4.

## A.3  EVALUATION METRICS

Information obtains from Fabbri et al. (2020):

- ROUGE measures the number of overlapping textual units between the generated summary and a set of reference summaries.
- ROUGE-WE extends ROUGE by taking cosine similarity of Word2Vec embeddings into account.
- BERTScore computes similarity scores by aligning generated and reference summaries on a token-level based on the output of the BERT-based model. Token alignments are computed greedily with the objective of maximizing the cosine similarity between contextualized token embeddings. We report the F1 score.
- MoverScore measures semantic distance between a summary and reference text by making use of the Word Mover's Distance operating over n-gram embeddings pooled from BERT representations.
- Sentence Mover's Similarity (SMS) extends Word Mover's Distance to view documents as a bag of sentence embeddings as well as a variation which represents documents as both a bag of sentences and a bag of words.
- BLEU is a corpus-level precision-focused metric which calculates n-gram overlap between a candidate and reference utterance and includes a brevity penalty. It is the primary evaluation metric for machine translation.
- CIDEr computes 1-4-gram co-occurrences between the candidate and reference texts, down-weighting common n-grams and calculating cosine similarity between the ngrams of the candidate and reference texts.

Table 6: Test set example for qualitative study.

| | |
|---|---|
| Phil: good evening deana ! many thanks for this nice card from you . constantine was very happy !. are these sunglasses also from you ? Deana: i thought they belonged your cathreen ! Phil: nope . she says they aren't hers . Deana: mine either . look , maybe you feel like keeping them ?. i seem to have so many sunglasses .. 8 Phil: where did you find them , possible that they belong to adrian ? Deana: in this empty place above the radio . in the very back .. if adrian wants it , no pro !. exactly ! Phil: ok , they don't belong to any of us , and nobody else drove your car . but we can look after these sunglasses . Deana: glad to hear it ! | |
| Longest-1 | Phil said good evening deana ! many thanks for this nice card from you . constantine was very happy !. are these sunglasses also from you ? |
| BART-1 | Phil and Deana will look after Adrian's sunglasses. |
| CORDIAL-1 | Deana found Adrian's sunglasses in the back of Phil's car. |
| BART | Phil and Deana are going to look after Adrian's sunglasses. |
| CorDial | Phil got a card from Deana. Deana found them in the empty place above the radio. Deana has a lot of them. |
| Gold | Phil received a card from Deana. Constantine was happy. Phil has sunglasses, that Deana found in the back above the radio. Deana and Phil don't know who they belong too. Phil will keep the sunglasses. |

Table 7: Test set example for qualitative study.

| | |
|---|---|
| Celia: where do you want to go for holiday ? Mike: i was thinking about egypt Celia: too hot . what about croatia ? Mike: good idea , i've never been there | |
| Longest-1 | Celia said where do you want to go for holiday ? |
| BART-1 | Mike wants to go for holiday to Egypt. |
| CORDIAL-1 | Mike wants to go on holiday to Egypt or Croatia. |
| BART | Celia and Mike will go for holiday to Croatia. |
| CorDial | Mike wants to go on holiday to Egypt. Celia thinks it's too hot. Mike has never been to Croatia, but he likes the idea. |
| Gold | Mike considers going to Egypt for holiday. It's too hot for Celia, she suggests Croatia instead. Mark likes the idea, he's never been there. |

Table 8: Test set example for qualitative study.

| | |
|---|---|
| Diane: how long do you have to work tonight ? Ross: about 2 hours , why ? Diane: i just wanted to do something maybe Ross: i think i'll be worn out after all hat work , baby Diane: we can just chill at home , don't worry. i just wanted to prepare Ross: ok. then just to be safe let's say it will take me 3 hours Diane: but you just said 2 ! Ross: d*mn it , Diane , don't start again Diane: what am i starting !. you're impossible Ross: can't you understand that this is important to me !. my career depends on it ! Diane: well , if your career is the most important thing in the world then i wouldn't want to disturb ! | |
| Longest-1 | Diane said well , if your career is the most important thing in the world then i wouldn't want to disturb ! |
| BART-1 | Ross has to work for 2 hours tonight. |
| CORDIAL-1 | Ross has to work 3 hours tonight. |
| BART | Ross has to work tonight for 2 hours. Ross and Diane will chill at home. |
| CorDial | Ross has to work 3 hours tonight. |
| Gold | Diane is not happy with Ross prioritising work over spending time with her. |

Table 9: Test set example for qualitative study.

| | |
|---|---|
| Finn: hey | |
| Zadie: hi there ! what's up ? | |
| Finn: all fine . you ? | |
| Zadie: not bad , thanks | |
| Finn: look , i was thinking of going to this neighborhood called elephant and castle tomorrow , it's apparently full of latin american stuff . fancy joining ? | |
| Zadie: sure ! but what's " stuff " ? | |
| Finn: lol so apparently it's a place were random people from " latin america " ( meaning fuck knows which countries ) started running small businesses and restaurant , and a nice little community was formed | |
| Zadie: oh cool | |
| Finn: then capitalism came and it's all going to be demolished soon , so it's like the last chance to go | |
| Zadie: what a shame yeah , i haven't had latin american food for ages so i'm totally up for it | |
| Finn: can't wait to taste this cuisine of unspecified latino origin lol. but we can specify time and place if and only if you wish | |
| Zadie: i might be tempted to lol i'd say early evening , 2 - ish ? | |
| Finn: yeah , that's fine by me . so most of the places we want to visit are in this elephant and castle shopping centre . shall i see you at the main entrance , wherever that is | |
| Zadie: 2 o'clock at unspecified main entrance then ? sounds good to mw | |
| Finn: yer | |
| Zadie: cool , see you there ! and thanks so much for remembering about me | |
| Finn: thanks for saying yes to such an ill-defined plan lmao | |
| Zadie: ha ha you know i love those | |
| Finn: see you tomorrow then | |
| Zadie: yep call me if you get lost | |
| Finn: i will i will bye | |
| Zadie: toodles | |
| Longest-1 | Finn said yeah , that's fine by me . so most of the places we want to visit are in this elephant and castle shopping centre. shall i see you at the main entrance , wherever that is |
| BART-1 | Finn and Zadie will meet tomorrow at 2 o'clock at the Elephant and Castle shopping centre. |
| CORDIAL-1 | Finn and Zadie are going to the Elephant and Castle tomorrow. |
| BART | Finn and Zadie are going to eat Latin American food tomorrow. They are meeting at the Elephant and Castle shopping centre at 2 pm. |
| CorDial | Finn and Zadie are going to visit a Latin American neighbourhood tomorrow. Zadie will call Finn if he gets lost. |
| Gold | Finn and Zadie are going to Elephant and Castle tomorrow at 2. They will meet at the main entrance. |

Table 10: Dialogue for the "Extract information after the discussion" sample in Table 5

| | |
|---|---|
| Paul: what color flowers should i get | |
| Cindy: any just not yellow | |
| Paul: ok , pink ? | |
| Cindy: no maybe red | |
| Paul: just tell me what color and what type ok ? | |
| Cindy: ugh , red roses ! | |
| Gold | Paul will buy red roses following Cindy's advice. |
| BART | Paul wants to get red roses. Cindy doesn't want pink or yellow roses. |
| CorDial | Paul wants to buy red roses. |

