# OpenReview forum: "CorDial: Coarse-to-fine Abstractive Dialogue Summarization with Controllable Granularity"
_ICLR.cc/2021/Conference — Reject_

### Official Review · AnonReviewer1 · 2020-10-27
**Review #1**

**Rating:** 4
**Confidence:** 5

**Review:**


<Summary>

This paper addresses the problem of abstractive dialogue summarization. Its key idea is to label an interrogative pronoun category and extract key phrases from each dialogue turn as weak guide for dialogue summarization. It also proposes a length-controllable generation method for final summary. The proposed approach is evaluated on the SAMSum as one of the largest abstractive dialogue summarization benchmarks, on which it shows competitive performance over recent models.

<Strengths>

1. It proposes a two-step coarse-to-fine approach for abstractive dialogue summarization; it first extracts category labels and key phrases from each dialogue turn, and then generates final summaries by controlling granularity. This idea itself could be novel.

2. It shows strong performance over other recent methods on the recently released SAMSum dataset.

3. It tests the proposed approach with four recent pre-trained language models including DialoGPT, UniLM, PEGASUS and BART.

<Weakness>

1. This paper proposes a novel coarse-to-fine approach for abstractive dialogue summarization but its implementation is largely ad-hoc and engineering intensive and thus bears little technical novelty.

(1) The “coarse” part aims at generating drafts using interrogative pronoun category prediction and key phrase  extraction.
These two are largely based on existing techniques (e.g. Ratner et al 2019 and Kitaev & Klien 2018) and some heuristics (e.g. thresholding for key phrases detection).

(2) The “fine” part aims at generating target summary with controllability of granularity.
Its implementation is also based on a series of engineering heuristics (e.g. dialogue splitting by ROUGE score, binary classification for cutpoint detection).

(3) In summary, it is hard to find methodological novelty in the proposed method.  Given that ICLR is a top premier ML venue, it could be a significant weakness to be a publishable work.

2. Experimental results are rather weak.

(1) Although SAMSum dataset may be one of the best benchmarks for the target task, experiments on only a single dataset is limited to show the generality and effectiveness of the proposed method.
Given that the proposed method is ad-hoc, I suspect much additional endeavor may be required to apply to another dataset.

(2) I am not sure whether the comparison in Table 1 is fair enough. Since the proposed approach relies on the additional components for draft construction, it could require more other types of training data or learned modules that other method may not need. This should be clarified in the draft.

<Conclusion>

My initial decision is ‘reject’ mainly due to lack of technical novelty. Limited experiments could be another issue to be improved.

---

> ### Author Response · Authors · 2020-11-16
> **Thank you so much for your review**
>
> Dear reviewer,
>
> Please let us answer your concerns or questions in the following:
>
> **[largely ad-hoc and engineering intensive]**
> Our work is more like an empirical work and the main contribution and goal is not to propose a new generative model or a new optimization algorithm, instead, we are trying to make a claim that for dialogue summarization task: 1) a noisy augmented draft can actually help final summary quality if an auto-regressive way, and 2) a simple strategy enables us to control summary granularity and it works surprisingly well. We are the first to propose a simple yet efficient training solution that surpasses any existing dialogue summarization models, including contemporaneous works [1][2]. We will include the reference you pointed out in our final version.
> [1] Multi-View Sequence-to-Sequence Models with Conversational Structure
> for Abstractive Dialogue Summarization (EMNLP 2020)
> [2] Improving Abstractive Dialogue Summarization with Conversational Structure and Factual Knowledge (under review, ICLR 2021)
>
> **[Experimental results on one dataset are rather weak]**
> SAMSum is the largest dialogue summarization dataset with high-quality annotations that we can evaluate on. We will include the AMI evaluation (a very small one) in the final version. Table 1 is a fair comparison because it can prove that our training strategy is useful and generalizable. Our method is not restricted to the BART model and it can be applied to any existing or future pre-trained language models.

---

### Official Review · AnonReviewer4 · 2020-10-28
**Correlation between the given problem and the proposed solution?**

**Rating:** 5
**Confidence:** 4

**Review:**

The paper proposes CorDial for abstractive dialogue summarization. CorDial extends BART by generating an intermediate "summary draft" which provides weakly-supervised signals and controling the length of the final summary. Results show significant improvements over competitive summarization models such as PEGASUS and BART in multiple different metrics.

Some comments:

1. The paper emphasizes that dialogue summarization is challenging due to its multi-speaker standpoints, casual language, and limited data. Although the use of the proposed summary draft would help solve the first challenge, it is hard to see any correlation between the other problems mentioned and the proposed solutions in the paper. This is especially the case for the controlling of the summary length. Why is this useful specifically for dialogue summarization?

2. The "summary draft" is one kind of a content plan, which is widely used in text generation, including text summarization [1]. The technique of extracting key phrase is similar to how content selection is done in [2]. Please compare the proposed solution to other kinds of content planning.

3. To extract key phrases, the method identifies the longest common sub-sequence (LCS) parameterized by a threshold, however how this threshold is set and used is not discussed in the paper. This is important information in order to understand how these key phrases would look like. For example, in Figure 1, how is "s just one of many boring days at work" extracted when the LCS is only "at work" for turn 2?

[1] https://www.aclweb.org/anthology/C18-1101.pdf

[2] https://arxiv.org/pdf/1808.10792.pdf

---

> ### Author Response · Authors · 2020-11-16
> **Thank you so much for your review**
>
> Dear reviewer,
>
> Please let us answer your concerns or questions in the following:
>
> **[Why Controllability of Dialogue Summarization?]**
> Controllable generation itself is an interesting yet difficult problem, and in this paper, we focus specifically on dialogue summarization because dialogue applications have received much attention due to the prevalence of smart speakers. We believe there is a vast application potential of dialogue summarization systems, and most importantly, it will be an appealing feature if we can control the granularity of summary based on user preference.
>
> **[Compared to Existing Draft Construction]**
> Our work is closely related to the literature of “retrieval-augmented generation”. The goal of our work is to show that even with a good pre-trained language model, we can still further improve its generation performance by fine-tuning it with weakly-annotated labels. What is the best way to construct a summary draft is not the main investigation of this work (Actually our solution is similar and more advanced to the Bottom-Up [2] approach). We will add one existing extractive-augmented method as a reference in the final version.
>
> **[Longest Common Subsequence (LCS)]**
> We do LCS after the parsing using the trained constituency parser (Section 2.2), so our LCS results are dependent on the parsing results. In this particular example, `s just one of many boring days at work`     the parsed constituent overlapping with ‘at work’ in the summary. However, in other examples, not all overlapped words are meaningful (e.g. stop words). We thus filter the LCS results and only keep important key phrases. We’ll explain more details about the process in the Appendix.

---

### Official Review · AnonReviewer2 · 2020-10-29
**Review for CorDial**

**Rating:** 5
**Confidence:** 4

**Review:**

This paper proposes CorDial, a new method for dialog summarization. CorDial firstly constructs a coarse draft by generating intent and key phrases for every dialog turn, and splits the dialog into chunks by inserting special boundary tokens; then the segmented original dialog and the constructed draft are feed as input to generate the final summary. CorDial employs BART-xsum as its backbone model, which is a pre-trained language model finetuned on XSUM summary dataset. Experiment result on SAMsum dataset shows CorDial achieves SotA performance under both automatic evaluation metric and human evaluations.

Overall, the paper presents an interesting, practical recipe for dialog summarization. It requires few additional annotation besides the summary, and the human evaluation results looks promising. However, the method proposed here somewhat lacks in novelty, and some part of the paper is not clearly written. Thus I give this paper a weak reject rating.

Comments:
1. In 2.2, how are the intents annotated? Is it a purely automatic process based on keywords matching? Or the keywords are merely cues for human annotators?
2. In 2.3, the algorithm for finding the cutting points is an incremental one. It can't account for the similarity between last chunk and the last sentence in the summary, since the cutting point of second to last chunk already determines the boundary of the last chunk.
3. How are the output generated exactly? The last sentence in 2.4 states each sentence is generated separately. Does it mean each output sentence have different input? How is it different from standard auto-regressive token-by-token generation?
4. 2.4 should also explicitly refer to Figure. 1 for clarity.
5. CorDial uses the BART-xsum as initialization, which is trained on XSUM dataset. Are other baselines also gone through the same XSUM training?
6. What is the model size of all the models in the experiments? It would be better to have some descriptions on model architectures in the experiment section.

---

> ### Author Response · Authors · 2020-11-16
> **Thank you so much for your review**
>
> Dear reviewer,
>
> We will try our best to make every part of the paper clear in the final version. Please let us answer your concerns or questions in the following:
>
> **[Novelty]**
> We propose a simple yet efficient solution that surpasses any existing dialogue summarization models, including contemporaneous works [1][2]. The main contribution of our work is not to propose a new generative model (we rely on pre-trained language models, which can be replaced or improved by any SOTA LMs), instead, we are trying to make a claim that 1) a noisy augmented draft can actually help final summary quality in a simple auto-regressive way, and 2) a simple strategy enables us to control summary granularity and it works surprisingly well.
> [1] Multi-View Sequence-to-Sequence Models with Conversational Structure
> for Abstractive Dialogue Summarization (EMNLP 2020)
> [2] Improving Abstractive Dialogue Summarization with Conversational Structure and Factual Knowledge (under review, ICLR 2021)
>
> **[How are the intents annotated?]**
> It is a totally automatic process. It is better than “keywords matching” since we use the Snorkel library to weakly label the data.
>
> **[Finding the last cutting points]**
> Yes, we have the assumption that the last chunk must be similar to the last summary sentence. How to further improve the mapping could be one important and interesting future work.
>
> **[How are the outputs generated?]**
> For CorDial, output summary is generated sentence-by-sentence, with the “same input” but different “highlighted portions” (Section 2.3). For example, <hl> X1, X2 <hl/> X3, X4, X5 → Y1 and X1, X2 <hl> X3, X4, X5 <hl/>  → Y2.
>
> **[XSUM for initialization]**
> The “BART” results shown in the paper are all “BART that is fine-tuned with XSUM data”. PEGASUS model itself is not pre-trained on XSUM but it is pre-trained by important sentence masking.
>
> **[Model Size]**
> For BART and CorDial, we are using a BART-large model with 400M parameters. PEGASUS has 568M parameters. We will include the model size information in the Appendix.

---

### Official Review · AnonReviewer3 · 2020-10-30
**This paper proposes CORDIAL to improve the abstractive dialogue summarization quality and controllability. It has a coarse-to- fine generation strategy that generates a summary draft followed by a final summary and a simple strategy to control the granularity of the final summary.**

**Rating:** 6
**Confidence:** 4

**Review:**

This paper is well written and investigates dialogue summarization that has not received much attention. It proposes a new model called CORDIAL which can generate a summary draft followed by a final summary.  It achieves comparable or better results in term of both automatic evaluation metrics, e.g. compress ratio, rouge score, and human evaluations (Consistent and Informative) about the quality of generated summaries in different settings.

However, there are still several disadvantages of this paper:
(1) It can generate a summary draft but its quality is almost not presented in the paper except the ablation study of table 1. Even the results within table 1 are still about the quality of the final summary. The paper slightly overclaims its usefulness in the draft summary generation. More results or analysis about draft summary should be presented.
(2) The human evaluation has only 30 examples and the scale is too small. Also, does the score is -1, 0, 1, or other scale? Do you use majority vote or mean plus standard deviation to get results in the table? Why gold in table 3 is so low in Consistent?
(3) The method looks applicable to the generable summarization task. More results of this are also interesting.

Although these drawbacks, its quality is good overall.

---

> ### Author Response · Authors · 2020-11-16
> **Thank you so much for your review**
>
>
> Dear reviewer,
>
> Please let us answer your concerns or questions in the following and we will incorporate all your feedback in the final version.
>
>  **[summary draft quality]**
> Good suggestion and we will include some generated drafts in the Appendix. So far, we treat the draft in our approach as an intermediate text, and what we care about the most is the quality of the final summary. Thus, it may not be that insightful to look at those “drafts” as they are constructed for weak supervision purposes (Section 2.2).
>
>  **[Human Evaluation Scale]**
> We use roughly 6% of the test set data for human evaluation and we do some filtering based on the annotation of the “gold summary”. Specifically, we filter those annotations if a “gold summary” has been annotated as “-1” (the meaning of each score is shown below), implying that the annotators may not pay attention to the scoring. The final results reported in Table 3 is the mean from three different annotators after filtering (4% data).
>
> The “gold summary” is actually not perfect and it might contain some noisy annotation, this is the reason why some workers may give 0 even if it is a “gold summary”. Please check the information below for the scoring instruction we sent to our workers. We will conduct one more round of human evaluation and include all of them in the final version:
>
> * Factual Consistency (Precision): The rating measures whether the information provided in a summary is correct.
> Score -1 if a summary contains a serious factual error.
> Score 0 if a summary has some minor factual errors.
> Score 1 if everything in a summary is factually correct.
>
> * Informative (Recall): The rating measures whether all the important information in a dialogue is included in a summary.
> Score -1 if a summary misses serious key points.
> Score 0 if a summary misses a few key points.
> Score 1 if a summary covers all key points.

---

### Decision · Program_Chairs · 2021-01-07
**Final Decision**

**Decision:**

Reject

**Comment:**

The paper proposes a method for the interesting task of dialog summarisation which is slowly getting attention from the research community. In particular, they propose a method which first generates a summary draft and then a final draft.

Pros:
1) The paper is well written
2) Addresses an interesting problem
3) SOTA results

Cons:
1) Lack of novelty
2) No quantitative analysis of the summary draft though it is as an important part of the proposed solution
3) Human evaluations are not adequate (the authors have said they will expand on this but clear details are not provided)
4) The BART model seems to have some advantage as it is pre-trained on XSUM data whereas some of the other models are not (the authors haven't clarified this sufficiently in the rebuttal)

Overall, the reviewers were not completely happy with the work and there was not clear champion.